Ecological and pest-management implications of sex differences in scarab landing patterns on grape vines

González-Chang Mauricio Mauricio.GonzalezChang@lincolnuni.ac.nz mauriciogonzalezchang@gmail.com 1
Boyer Stéphane 2
Lefort Marie-Caroline 1 2
Nboyine Jerry 1
Wratten Steve D. 1
1 Bio-Protection Research Centre, Lincoln University , Christchurch , Lincoln , New Zealand
2 Environmental and Animal Sciences, Unitec Institute of Technology , Auckland , New Zealand
Huber Dezene
Electronic publication date: 2017 Apr 27
Publication date: 2017
Volume: 5
Electronic Location ID: e3213
Received 2016 Nov 21; Accepted 2017 Mar 21
Copyright: ©2017 González-Chang et al.
Copyright year: 2017
Copyright holder: González-Chang et al.
License: This is an open access article distributed under the terms of the Creative Commons Attribution License, which permits unrestricted use, distribution, reproduction and adaptation in any medium and for any purpose provided that it is properly attributed. For attribution, the original author(s), title, publication source (PeerJ) and either DOI or URL of the article must be cited.
License URL: https://creativecommons.org/licenses/by/4.0/

Keywords: Melolonthinae, Sex-ratio, Landing behaviour, Vineyards, Generalised linear models

Funding: New Zealand Winegrowers Callaghan Innovation Scholarship KNZLP1201 Kono Beverages The New Zealand Winegrowers partly funded the 2015 sampling season. This research was done as part of M.G. Chang’s PhD programme, funded by a Callaghan Innovation Scholarship (KNZLP1201) and Kono Beverages. The funders had no role in study design, data collection and analysis, decision to publish, or preparation of the manuscript.

==============================
Background

Melolonthinae beetles, comprising different white grub species, are a globally-distributed pest group. Their larvae feed on roots of several crop and forestry species, and adults can cause severe defoliation. In New Zealand, the endemic scarab pest Costelytra zealandica (White) causes severe defoliation on different horticultural crops, including grape vines (Vitis vinifera). Understanding flight and landing behaviours of this pest can help inform pest management decisions.

Methods

Adult beetles were counted and then removed from 96 grape vine plants from 21:30 until 23:00 h, every day from October 26 until December 2, during 2014 and 2015. Also, adults were removed from the grape vine foliage at dusk 5, 10, 15, 20 and 25 min after flight started on 2015. Statistical analyses were performed using generalised linear models with a beta-binomial distribution to analyse proportions and with a negative binomial distribution for beetle abundance.

Results

By analysing C. zealandica sex ratios during its entire flight season, it is clear that the proportion of males is higher at the beginning of the season, gradually declining towards its end. When adults were successively removed from the grape vines at 5-min intervals after flight activity begun, the mean proportion of males ranged from 6–28%. The male proportion suggests males were attracted to females that had already landed on grape vines, probably through pheromone release.

Discussion

The seasonal and daily changes in adult C. zealandica sex ratio throughout its flight season are presented for the first time. Although seasonal changes in sex ratio have been reported for other melolonthines, changes during their daily flight activity have not been analysed so far. Sex-ratio changes can have important consequences for the management of this pest species, and possibly for other melolonthines, as it has been previously suggested that C. zealandica females land on plants that produce a silhouette against the sky. Therefore, long-term management might evaluate the effect of different plant heights and architecture on female melolonthine landing patterns, with consequences for male distribution, and subsequently overall damage within horticultural areas.

Introduction

White grubs (Coleoptera: Scarabaeidae: Melolonthinae) are a widely distributed group of herbivorous insects feeding on a variety of plant hosts around the world (Jackson & Klein, 2006). Their larvae feed on plant roots, leading to severe damage to commercial crops and pastures (Jackson & Klein, 2006; Frew et al., 2016), as well as to natural forests and forestry plantations (Švestka, 2006; Švestka, 2010). Some melolonthines do not feed as adults, like Leucopholis lepidophora (Blanchard) (Kalleshwaraswamy et al., 2016) and Phytholaema herrmanni (Germain) (Durán, 1954), while others cause minor plant defoliation. These include Schizonyza ruficollis (F.) (Kulkarni et al., 2007) and Holotrichia spp. (Kulkarni et al., 2009) in India, Hoplia philanthus (Füessly) in Belgium (Ansari et al., 2006), and Phyllophaga cuyabana (Moser) in Brazil (Oliveira & García, 2003). However, dramatic defoliation can also occur. In Eastern Asia, Ectinohoplia rufipes (Motschulsky) adults can severely defoliate ornamental trees around golf courses, as well as in gardens and parks (Kim et al., 2013). In Central and Western Europe, adult defoliation by Melolontha melolontha (L.) and Melolontha hippocastani (F.) has led to severe defoliation on different horticultural crops, such as grape vines (Vitis vinifera (L.)) and forest trees, respectively (Jackson & Klein, 2006; Reinecke et al., 2006; Švestka, 2006; Švestka, 2010). Such defoliation also has been recorded in New Zealand, where Costelytra zealandica (White) attacks several horticultural crops, including tamarillo (Solanum betaceum (Cav.)), avocado (Persea americana (Mill.)), blueberries (Vaccinum corymbosum (L.), strawberries (Fragaria x ananassa (Duchesne)), kiwifruit (Actinidia chinensis (Planch)), and grape vines (Binfield, 1933; Blank & Olson, 1982; Blank, Olson & Bell, 1983; East, Willoughby & Koller, 1983; East & Holland, 1984). Sometimes, severe defoliation can occur (Blank, Olson & Bell, 1983; East, Willoughby & Koller, 1983; Blank, 1992), leading in some cases to prophylactic application of synthetic insecticides to reduce C. zealandica damage. Nowadays, the use of such an approach is discouraged due to the environmental and human health problems related to their use (Reganold & Wachter, 2016).

The importance of the defoliation produced by melolonthines has promoted investigation of their plant colonization patterns, subsequently leading to work on their flying and landing behaviour. In Melolonthinae, after landing on their host plants, females attract males by releasing sex pheromones (Henzell & Lowe, 1970) or by the release of green leaf volatiles (GLV) produced after feeding on plant foliage (Harari, Ben-Yakir & Rosen, 1994; Reinecke, Ruther & Hilker, 2002a), or both acting together (Reinecke et al., 2002b; Reinecke et al., 2006). However, in Maladera matrida (Argaman), males land on their plant hosts before females. By feeding on their host plant, GLV are released attracting the females to where feeding occurs (Harari, Ben-Yakir & Rosen, 1994). Observations on C. zealandica flying behaviour have suggested that females land before males on shrubs and trees (Farrell & Wightman, 1972). After landing, males are attracted to those plants by the female pheromone, identified as phenol (Henzell & Lowe, 1970). Although the effects of phenol on male attraction are well understood for C. zealandica (Henzell & Lowe, 1970; Unelius et al., 2008; Marshall et al., 2016), M. melolontha, M. hippocastani (Reinecke et al., 2006), and P. cuyabana (Zarbin et al., 2007), there are no studies quantifying adult C. zealandica sex ratio at the landing phase of its daily plant colonising activity. In addition, no research has reported the trends in sex ratio throughout the entire C. zealandica flight season for any agricultural or natural system. We hypothesized that females arrive on V. vinifera before males. Therefore, the aims of this work were: (i) to quantify C. zealandica sex ratio throughout its seasonal and daily flight period; and (ii) to correlate adult abundance with sex ratio through its seasonal and daily flight activity.

Material and Methods

Study sites

This work was conducted in the Marlborough region of New Zealand. In this area, two commercial vineyards were chosen. One in the Awatere Valley (41°44′S; 173°52′E) owned by Kono Beverages, and another close to Blenheim city (41°33′S; 173°55′E) that belongs to Wither Hills. These locations are situated in homogeneous landscapes generally dominated by conventionally-managed vineyards. To reduce the effect of conventional management practices on the interpretation of the observations described below, organically-managed vineyard blocks (cv. Pinot Noir) were chosen at both sites. The sizes of the blocks were 6.12 and 4.58 ha in the Awatere Valley and Blenheim, respectively. Grape vines were 14 and 15 yr old in the Awatere Valley and Blenheim, respectively. Plant spread along the vine row was 1.8 m between each other. During the study period, from late October to late November 2014 and 2015, no pesticides or herbicides were applied on those blocks, apart from sulphur, which was applied to control fungal diseases. In New Zealand, organic vineyard areas, including headlands, inter-row and under-vine areas are covered by a mixture of ground covers that mainly comprise ryegrass (Lollium perenne (L.)), white clover (Trifolium repens (L.)), and fescue (Festuca spp.).

Adult C. zealandica sampling

Sampling was carried out every day during C. zealandica flight season. In the Marlborough area, individuals start flying from the end of October until the end of November (Farrell & Wightman, 1972). Within the October to November period, adults initiate flight 20 min after sunset, flying for approximately 26 min (M González-Chang, 2016, unpublished data). During their flight activity, adults land on grape vine foliage and then feed and mate, staying on the grape vines for at least 3 h (M González-Chang, 2016, unpublished data). Based on this landing behaviour, adults were counted and then removed from 96 plants at both study locations from 21:30 until 23:00 h. These plants had a mean height of 1.5 m, and were consecutively sampled along the vine row over three consecutive rows. Sampling started along the row at the edge of a vineyard block to 50 m from the edge, sampling 32 plants on each row. The row sampling method was performed at the Awatere Valley during 2014 and 2015 and in Blenheim only on 2015.

Seasonal sex ratio

Daily during the 2014 flight season from October 27th until November 30th, beetles from the Awatere Valley were collected by hand. During the 2015 flight season, beetle collection started in the Awatere Valley from November 7th until December 1st, while in Blenheim from October 26th until December 1st. Individual polyethylene bags were used to collect the beetles, so samples were not mixed between days or locations. Using a 50 ml cylindrical plastic container, a sub-sample was taken from each bag for sex ratio analysis. Within this sub-sample the ratio was analysed by randomly sexing 20 individuals. This procedure was repeated three times for each day. The proportion of males (M) over females (F) for each sampled day was obtained after taking a mean of the three measurements. This proportion was calculated as M/(M+F). Days without flight activity were removed from further sex-ratio analysis. The separation of males from females was carried out using the morphological characteristics proposed by Kain (1972) and Kelsey (1965). Using a stereomicroscope (Carl Zeiss), the distal-ventral section of the abdomen was examined for male characters, including a shallow depression in the centre of the sixth abdominal sternite (Kelsey, 1965), and the presence of parameres through the pygdium (Kain, 1972). Female characters included the presence of colleterial glands, as two spherical dots in the fifth sternite, and two genital sclerites situated at each side of the vulva (Kain, 1972).

Adult C. zealandica removal to determine daily sex-ratio patterns

To investigate the dynamics of sex-ratio changes on grape vine foliage during the flight season, adult beetles were successively removed from the grape vines at different periods during 2015. At both sites, adults were removed from the grape vine foliage at dusk at 5, 10, 15, 20 and 25 min after flight started. One plant at the edge of each vineyard block at both locations was used for this experiment. This plant was not included in the 96 plants used in the seasonal sex-ratio determination explained above. Adults were visually counted on the selected grape vine and then removed by hand at each time period. When the number of adults collected exceeded 60, 20 individuals were sub-sampled three times from those samples. The mean of the sub-sampled was used for further statistical analyses. This daily sequential adult removal was performed from November 2 to 28 and November 14 to 28 at Blenheim and the Awatere Valley, respectively. Days without flight activity, because of adverse weather, were removed from further sex-ratio analysis. The mean proportion of males and total adult numbers at each sampling day at each time interval were considered as replicates.

Statistical analyses

The proportions obtained for each day during the sampling season, and those calculated at each evaluated time period, were analysed using a generalised linear model (GLM), with a beta-binomial distribution and “logit” as the link function (Cribari-Neto & Zeileis, 2010). It has been previously suggested that the beta regression approach is inherently heteroscedastic and easily accommodates asymmetries typically found in rates and proportions (Cribari-Neto & Zeileis, 2010; Grün, Kosmidis & Zeileis, 2012). Beta-binomial GLMs were calculated with the R package “betareg” (Cribari-Neto & Zeileis, 2010). Spearman’s rank correlation was used to assess the correlation between adult abundance and sex ratio through the flight season (Crawley, 2007). The same analysis was used to evaluate the correlation between adult abundance and the sex ratio at each time period studied. Differences in adult abundance between periods were analysed using Tukey’s multiple contrasts post-hoc analysis (Venables & Ripley, 2002) after fitting a GLM with a negative binomial distribution and a logarithmic link function at each sampling site. Negative binomial GLM and Tukey’s tests were performed using the R packages “MASS” and “multcomp”, respectively. All statistical analyses were performed using the statistical software R v.3.2.5 (R Core Team, 2016).

Figure 1 Costelytra zealandica sex-ratio (proportion of males) on 35 d after its flight begun, from October 27th to November 30th at the Awatere Valley in 2014.

The beta-binomial GLM (solid line) adjusted to each proportion during the flight season is presented with its respective confidence intervals (dashed lines). The proportion of males is reduced through time (slope =  − 0.08; z =  − 4.8; p < 0.001). Days without flight activity during the season were removed from this analysis.

Results

Seasonal sex ratio

In the Awatere Valley in 2014, a total of 36,369 adults were sampled, while in 2015 only 6,111 were collected. During 2015 in Blenheim, 14,731 adults were collected. Sex-ratio was not statistically correlated with beetle abundance through the flight season in the Awatere Valley during 2014 (φ = 0.02; p = 0.92) nor in 2015 (φ =  − 0.35; p = 0.12). However, at the Blenheim site, this correlation was positive and significant (φ = 0.51; p < 0.01). In the Awatere Valley the proportion of males decreased through time during 2014 (slope =  − 0.08; z =  − 4.8; p < 0.001) and in 2015 (slope =  − 0.02; z =  − 2.6; p < 0.01). A similar result was observed in Blenheim during 2015 (slope =  − 0.03; z =  − 4; p < 0.001). Seasonal sex-ratio results are shown in Figs. 1–3.

Figure 2 Costelytra zealandica sex-ratio (proportion of males) on 25 d after its flight begun, from November 7th to December 1st at the Awatere Valley in 2015.

The beta-binomial GLM (solid line) adjusted to each proportion during the flight season is presented with its respective confidence intervals (dashed lines). The proportion of males is reduced through time (slope =  − 0.02; z =  − 2.6; p < 0.01). Days without flight activity during the season were removed from this analysis.

Figure 3 Costelytra zealandica sex-ratio (proportion of males) on 37 d after its flight begun, from October 26th to December 1st close to Blenheim city in 2015.

The beta-binomial GLM (solid line) adjusted to each proportion during the flight season is presented with its respective confidence intervals (dashed lines). The proportion of males is reduced through time (slope =  − 0.03; z =  − 4; p < 0.001). Days without flight activity during the season were removed from this analysis.

Adult C. zealandica removal to determine daily sex-ratio patterns

When adults were removed from the grape vine foliage every 5 min after flight activity started, the proportion of males increased through the 5–25 min time interval at Blenheim (slope = 0.06; z = 2.12; p < 0.05), but not at the Awatere Valley (slope = 0.001; z = 0.03; p = 0.98) (Fig. 4). Tukey contrasts showed that adult numbers significantly increased between 10 and 15 min after flight occurred at both sites, and then decreased to initial levels between 20 and 25 min after it started (Fig. 5). In the Awatere Valley, flight activity completely ceased 25 min after begun, while in Blenheim it stopped after 30 min. No correlation was found between adult abundance and the sex-ratio during each time period evaluated at Blenheim (φ =  − 0.1; p = 0.87) or in the Awatere Valley (φ =  − 0.8; p = 0.2).

Discussion

In this work, seasonal and daily changes in adult C. zealandica sex-ratio throughout its flight season are presented for the first time. The proportion of males significantly decreased towards the end of the season at both studied locations. However, when adults were removed from the grape vine foliage at several time periods during the same day, trends in the sex-ratio differed between the two locations. These results are discussed in terms of this insect’s behaviour. The implications of these results on sustainable C. zealandica control in vineyards also are discussed.

Figure 4 Proportion of C. zealandica males, after adult removal from a single grape vine plant at 5 min periods after daily flight activity begun, at the Awatere Valley (A) and Blenheim (B).

The beta-binomial GLM (solid line) adjusted to each proportion during those time periods is presented with its respective confidence intervals (dashed lines). In the Awatere Valley (right), male proportion did not change during C. zealandica daily flight activity (slope = 0.001; z = 0.03; p = 0.98), but in Blenheim (left), it increased through the daily flight time period (slope = 0.06; z = 2.12; p < 0.05). Error bars are two-standard errors.

Seasonal sex-ratio trend

A widely-accepted sex-ratio theory based on natural selection proposed by Fisher (1930), suggests that sex-ratio is equilibrated on a 1:1 proportion to promote species survival. However, many species do not follow the 1:1 theoretical proportion as they respond to their changing local environmental conditions (Hamilton, 1967). In Melolonthinae sex-ratio though their flight season can be female-biased (Méndez-Aguilar et al., 2005; Kalleshwaraswamy et al., 2016), male-biased (Ansari et al., 2006; Švestka, 2006; Švestka, 2010) or at a 1:1 ratio (Harari, Ben-Yakir & Rosen, 1994; Méndez-Aguilar et al., 2005). However, it is worthwhile noting that sex-ratio can vary between those proportions every day during the flight season (Švestka, 2006; Švestka, 2010; Kalleshwaraswamy et al., 2016), as shown in this study (Figs. 1–3). In this work, seasonal sex-ratio varied between years and locations, being male-biased (64% males) and female-biased (47% males), in the Awatere Valley in 2014 and 2015, respectively. In Blenheim, the seasonal sex-ratio was 1:1 (50% males). The sex-ratio differences between sites in this study highlights the temporal and spatial variability of melolonthine sex-ratio, which might be determined by environmental conditions (Sánchez, 2008; Gempe & Beye, 2011). Previous studies suggested that C. zealandica males appear seasonally before females do (Kelsey, 1951; Pottinger, 1968; Farrell & Wightman, 1972). Early male appearance might have important evolutionary consequences for the survival of C. zealandica because as soon as the females emerge, males are already present for mating (Kelsey, 1951; Durán, 1954; Harari, Ben-Yakir & Rosen, 1994; Kalleshwaraswamy et al., 2016). The evolutionary advantage of early male emergence is in agreement with the results presented in this work, as a higher proportion of males were found at the beginning of the flight season at both studied years and locations. The reduction in male numbers through time might be associated with a reduction in female pheromone production towards the end of the flight season, as pheromone effects on male attractiveness have been well studied in C. zealandica (Henzell, 1970; Unelius et al., 2008; Marshall et al., 2016). Recently, it has been suggested that phenol, the main sex pheromone in C. zealandica (Henzell, 1970), is produced by a mutualistic bacterium Morganella morganii (Brenner) in its colleterial gland, which is biosynthesized from the amino acid tyrosine (Marshall et al., 2016). In general, insect protein synthesis decreases with aging (Levenbook, 1986), and tyrosine in particular has been shown to decrease with age in other insects like Bombyx mori (L.) (Lepidoptera: Bombycidae) (Osanai & Kikuta, 1981). Considering that the C. zealandica flight season lasted for around 4 wk, from late October to late November at the studied vineyards, females that landed on the grape vine foliage at the end of that period might already have ceased to synthesize their pheromone due to aging, reducing the number of males attracted to them.

Figure 5 Adult abundance during C. zealandica daily flight activity at different 5-min time periods, for beetles that landed on a single grape vine plant at the Awatere Valley (dashed line) and Blenheim (solid line) sites.

Tukey contrasts are presented as different letters above each dot. Different letters represent significantly differences among the different time periods, but not between studied locations. Error bars are two-standard errors.

In the Awatere Valley, no correlation was found between adult numbers throughout the flight season and changes in the sex-ratio. However, in Blenheim, the correlation was positive and significant. A positive correlation suggests that an increase in adult numbers is due an increase in the number of males, because in the Awatere Valley 2014 flight season, 63% of the sampled adults at their November 14 peak in abundance (6,968 individuals) were males. Conversely, at the same location during 2015, only 37% of the adults were males at their November 19 peak (1,039 individuals). In Blenheim, 55% of them were males at their November 6 peak (1,966 individuals).

Adult removal and daily sex-ratio trends

Costelytra zealandica flight activity has been divided in two phases (Kelsey, 1951; Pottinger, 1968; Farrell & Wightman, 1972). Firstly, males emerge from the ground and hover over the grass. When females emerge, they climb to the top of a grass blade and release their sex pheromone to attract males. Once mating has occurred, females return to the ground to lay eggs. Secondly, females fly from the grass, searching for silhouettes in the sky, landing on trees, hedgerows and shrubs (Farrell & Wightman, 1972). After landing occurs, pheromones are released, attracting males from the surrounding areas again. Then, males and females drop to the ground and females again lay eggs, although in fewer numbers compared to their first ovisposition made close to their initial emergence sites (Farrell & Wightman, 1972). The results presented in this work are in agreement with the colonization sex-driven dynamics for the second flight phase observed during 1970s (Farrell & Wightman, 1972), in which females land first on the vegetation; as the proportion of males found when adults were removed from the grape vines never exceed 28%. In fact, 10-min after flight begun, only 9 and 18% of the removed adults were males at the Awatere Valley and Blenheim, respectively. Remarkably, an identical two-phase flight behaviour has been described before for Phyllopertha horticola (L.) (Coleoptera: Rutelinae) (Schneider, 1962) and for P. herrmanni (Durán, 1954). Also, similar landing behaviour where female melolonthines arrive before males on plant foliage has been reported for M. melolontha, M. hippocastani (Reinecke, Ruther & Hilker, 2002a) and P. cuyabana (Oliveira & García, 2003). In the Awatere Valley (2015), the proportion of males did not change throughout their daily flight activity, which lasted for 25 min. However, in Blenheim (2015) male proportion increased throughout their daily flight period, which totally ceased 30 min after flight begun. These differences in adult abundance and flight length at dusk could be related to the two-fold increase in adult numbers found in Blenheim compared to the Awatere Valley.

Female behaviour and plant silhouettes

Considering that males are attracted to females that have already landed on their host plant, through pheromone communication (Henzell & Lowe, 1970) and/or plant volatiles (Reinecke, Ruther & Hilker, 2002a), the mechanisms by which females locate host plants have relevance to pest management. Different authors have proposed that female host finding in Melolonthinae is through the visual recognition of the tallest plant silhouettes in the sky (Durán, 1954; Schneider, 1962; Farrell & Wightman, 1972; Oliveira & García, 2003). A study in Brazil showed that P. cuyabana females preferred tall soybean and corn plants for landing and mating (García, Oliveira & de Oliveira, 2003). Recent work suggests that plant location by C. zealandica females might be mediated by the contrast that plant silhouettes produce in the sky (M González-Chang, 2016, unpublished data). Males also might use visual cues in the sky for orientation at some extent, but more research is needed, as no evidence for the latter exist so far. Thus, additional research investigating the addition of tall plant species at the edge of crops could have management values, especially if the tall plants can divert females from landing further into the crop area, and in turn, deterring males from the crop.

Implications for sustainable C. zealandica management in vineyards

Recently, it has been suggested that a “push-pull” strategy (Khan et al., 2011) might contribute to the control of white grub populations (Dynastinae, Rutelinae, Melolonthinae) around the world (Frew et al., 2016). In the context of C. zealandica pest control in grape vines, it seems straightforward to use the sex attractant pheromone (phenol) to “pull” the adults away from the grape vines. Phenol, combined with another management intervention which “pushes” the adults away, as recently shown when crushed mussel shells were applied on the under-vine vineyard areas (M González-Chang, 2016, unpublished data), might arise as a novel variant on the “push-pull” approach to control this New Zealand pest in several horticultural crops. However, the use of phenol in such a way has some shortcomings. Although phenol was identified as the pheromone for C. zealandica more than 40 years ago (Henzell & Lowe, 1970), it has not been widely used to control this pest due constraints in trap design that led to low capture efficiency (Unelius et al., 2008). Another negative factor of using phenol is its high toxicity for humans (Shadnia & Wright, 2008; Unelius et al., 2008). Instead, if females can be attracted to a host plant outside the crop, the overall numbers of adult C. zealandica landing on the crop could be reduced, along with subsequent reduction in plant damage (M González-Chang, 2016, unpublished data). The mechanisms behind female C. zealandica host location remain unknown with literature only suggesting that females might locate host plants by their contrast against the sky at dusk (Farrell & Wightman, 1972). Despite those assumptions, it cannot be ignored that females can also be attracted to different floral plant volatiles, as demonstrated for P. horticola (Ruther, 2004). Furthermore, the potential contribution of GLV, released from adult feeding (Reinecke et al., 2002b), may influence seasonal C. zealandica peak activity and abundance, because peak damage coincides with peak flight activity in the middle of the flight season (East, Willoughby & Koller, 1983). In this work, the peak of abundance was registered during the middle of the flight season. Thus, the chemical ecology underlying C. zealandica reproductive behaviour needs further investigation. Another approach could evaluate the contribution of native New Zealand vegetation within or outside the vineyard (Shields et al., 2016), as non-crop vegetation could provide visual silhouettes and organic volatiles to further attract males or females. Although the establishment and growth of those native plants in horticultural areas will take time, native plants could contribute to a long-term sustainable C. zealandica pest management strategy, with potential enhancement of local functional biodiversity and biological conservation that need to be further addressed (Isaacs et al., 2009).

Conclusions

The proportion of C. zealandica males landing on the grape vine foliage decreased throughout the flight season during 2014 and 2015 at two New Zealand locations. Blenheim (2015) was the only site where increases in sex-ratio (male-biased) were correlated with increases in overall adult abundance. The sex-ratio correlation with adult abundance might be related to spatial and temporal landscape variations in sex-ratio between Blenheim and the Awatere Valley, although landscape factors were not measured here. When adults were removed daily from the grape vines at several time periods after their flight activity begun, a female-biased sex-ratio was recorded. The female-based sex-ratio suggests that females land on the plant foliage before males, subsequently attracting them by the release of their phenol sex pheromone. The grape vine colonization sex-driven dynamics during adult C. zealandica flight activity presented in this work might contribute to management useful in reducing the damage of this endemic New Zealand pest on several horticultural crops, and eventually in other scarabaeids with similar flight behaviour around the world.

Supplemental Information

Data S1 Raw dataset used for statistical analyses

All data needed for statistical computations

Click here for additional data file.

Supplemental Information 1 R code used to perform all the statistical analyses presented in the manuscript

This R code script presents the steps carried out to analyse the data used for this manuscript. For each analysis, individual datasets extracted from the raw data file were created and used as .txt files.

Click here for additional data file.

Supplemental Information 2 Text (.txt) file used for the R code

Data used to calculate the relationship between adult C ostelytra zealandica sex ratio and days after adults’ flight started in the Awatere Valley during the 2014 flight season. Data are shown in Fig. 1.

Click here for additional data file.

Supplemental Information 3 Text (.txt) file used for the R code

Data used to calculate the relationship between adult Co stelytra zealandica sex ratio and days after adults’ flight started in the Awatere Valley during the 2015 flight season. Data are shown in Fig. 2.

Click here for additional data file.

Supplemental Information 4 Text (.txt) file used for the R code

Data used to calculate the relationship between adult C ostelytra zealandica sex ratio and days after adults’ flight started in Blenheim during the 2015 flight season. Data are shown in Fig. 3.

Click here for additional data file.

Supplemental Information 5 Text (.txt) file used for the R code

Data used to calculate the relationship between adult C ostelytra zealandica sex ratio and daily adult removal from grape vines at different time periods in the Awatere Valley during the 2015 flight season. Data are shown in Fig. 4A.

Click here for additional data file.

Supplemental Information 6 Text (.txt) file used for the R code

Data used to calculate the relationship between adult C ostelytra zealandica sex ratio and daily adult removal from grape vines at different time periods in Blenheim during the 2015 flight season. Data are shown in Fig. 4B.

Click here for additional data file.

Supplemental Information 7 Text (.txt) file used for the R code

Data used to calculate the correlation between adult C ostelytra zealandica sex ratio and adult abundance in the Awatere Valley during the 2014 flight season.

Click here for additional data file.

Supplemental Information 8 Text (.txt) file used for the R code

Data used to calculate the correlation between adult C ostelytra zealandica sex ratio and adult abundance in the Awatere Valley during the 2015 flight season.

Click here for additional data file.

Supplemental Information 9 Text (.txt) file used for the R code

Data used to calculate the correlation between adult C ostelytra zealandica sex ratio and adult abundance in Blenheim during the 2015 flight season.

Click here for additional data file.

Supplemental Information 10 Text (.txt) file used for the R code

Data used to calculate the correlation between adult C ostelytra zealandica sex ratio and daily adult removal abundance from grape vines at different time periods in Blenheim during the 2015 flight season.

Click here for additional data file.

Supplemental Information 11 Text (.txt) file used for the R code

Data used to calculate the correlation between adult C ostelytra zealandica sex ratio and daily adult removal abundance from grape vines at different time periods in the Awatere Valley during the 2015 flight season.

Click here for additional data file.

Supplemental Information 12 Text (.txt) file used for the R code

Data used to calculate the mean adult C ostelytra zealandica abundance at 5-min time periods after adult flight activity begun in Blenheim during the 2015 flight season. Data are shown in Fig. 5 (solid line).

Click here for additional data file.

Supplemental Information 13 Text (.txt) file used for the R code

Data used to calculate the mean adult Costelytra zealandica abundance at 5-min time periods after adult flight activity begun in the Awatere Valley during the 2015 flight season. Data are shown in Fig. 5 (dashed line).

Click here for additional data file.

Supplemental Information 14 Text (.txt) file used for the R code

Data used to calculate Tukey’s contrasts between adult Costelytra zealandica abundance and the 5-min time periods after adult flight activity begun in Blenheim during the 2015 flight season. Data are shown in Fig. 5 (solid line).

Click here for additional data file.

Supplemental Information 15 Text (.txt) file used for the R code

Data used to calculate Tukey’s contrasts between adult Costelytra zealandica abundance and the 5-min time periods after adult flight activity begun in the Awatere Valley during the 2015 flight season. Data are shown in Fig. 5 (dashed line).

Click here for additional data file.

We gratefully acknowledge the assistance provided by Coralline Houise, Estelle Postic, Héléna Minet and Lucie Archard.

Additional Information and Declarations

Competing Interests

Author Contributions

Data Availability

Author Steve D. Wratten is an Academic Editor for PeerJ. The authors declare there are no competing interests.

Mauricio González-Chang conceived and designed the experiments, performed the experiments, analyzed the data, wrote the paper, prepared figures and/or tables.

Stéphane Boyer and Marie-Caroline Lefort wrote the paper.

Jerry Nboyine performed the experiments, wrote the paper, reviewed drafts of the paper.

Steve D. Wratten conceived and designed the experiments, contributed reagents/materials/analysis tools, wrote the paper, prepared figures and/or tables, reviewed drafts of the paper.

The following information was supplied regarding data availability:

The raw data has been supplied as a Supplementary File.

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
