# Peer review of "Ecological and pest-management implications of sex differences in scarab landing patterns on grape vines"

_PeerJ, doi:10.7717/peerj.3213_

## Round 0.1 · original submission · Major Revisions

As noted by Reviewer 1 and Reviewer 3, this MS seems mainly robust, well-written, and should provide important new information to the specific field (IPM in vineyards) and perhaps to the larger discipline of entomological IPM.

However the reviewers, particularly Reviewer 1 and 2, also noted deficiencies in the methodological descriptions. Those are important points and those issues need to be rectified and assessed prior to publication.

Reviewer 1 ·

Basic reporting

All of my comments are provided directly on the manuscript in track changes format. However, to address some of the PeerJ Review Form Sections:

1) Basic Reporting: My biggest concern with the manuscript was clarity and language usage. Many of the sentences were quite difficult to understand as written or were missing critical information. I tried to modify some of these where I could, but in other cases where it was unclear what the authors were trying to convey, I just indicated unclear and needs work. The author’s use of “this”, “these”, “it”, “that” in many places left it very unclear what they were referring too. I have marked most of these places and also provided suggestions for improving. In general, I had a hard time overall following the manuscript concepts and flow because I was sort of lost in the details of trying to understand each sentence. At times, I was quite confused in the text. Some of the suggestions I have provided may help. I would strongly suggest that the authors use “exactly” the same section headings in the Materials and Methods, Results, and Discussion sections so the reader knows how these sections relate. I am not sure it is necessary to have section headings throughout the discussion.
The authors did do a good job of providing a literature background to support the manuscript. They have adequate citations to support their concepts and have related them well to the study purpose. The sections that were most clear to me were the Abstract, Introduction, and Conclusions. The methods, results, and discussion were probably the most confusing and difficult to follow sections, especially the methods and discussion sections.
The structure seems to conform to PeerJ standards. I have modified some of the references to match PeerJ formats. Figures are appropriate and raw data was supplied. There were a lot of headings in the raw data file and it was not exactly clear to me how the raw data related to the summarized findings in the manuscript.

Experimental design

Research appears to be within scope of journal based on other articles I looked at in the journal. The research question was definitely relevant to the management of this pest and the research did fill in some missing knowledge gaps. As far as I can tell, the research was performed to high technical and ethical standard, but there definitely are not enough details and information in the methods to replicate the study. Some areas of the methods need greater clarity, including some of the missing information that I have indicated above and in the manuscript with regards to how sampling was performed. I think the conclusions section was probably one of the best written sections of the manuscript and the authors did a good job in the conclusions with linking to the original research question and supporting results. The authors did add some speculations in their discussion, but supported their assertions with other citations.

Validity of the findings

The research preformed addresses novel aspects of sex ratio and abundance patterns of an important scarab pest of multiple horticultural crops. The authors also related the potential impact of their findings to management needs and proposed ideas for how their work could be used in management (e.g., tall silhouettes to draw females away from crops and in turn also remove males that are attracted to female pheromones). The authors used several statistical procedures that were well referenced and supported. I think some improvements are needed in making the methods more clear. I did have a difficult time understanding how sampling was performed and the descriptions about 50 meters from field center were not helpful or descriptive, especially since field size would affect sample location between sites as explained. No descriptions were given about how the authors selected plants they sampled within rows, so the reader is left assuming it was somehow done randomly. They also said they used three rows, but did not explain if they were edge rows, consecutive rows, etc. All this needs more clarity since it affects the validity of the statistical testing and also the repeatability by other researchers. I have indicated on the manuscript in track changes most places where there was confusion to me in the methods and results sections. One topic I noticed the researchers did not address in the manuscript was how their removal of beetles in some of their tests might have affected subsequent response of other beetles.

Additional comments

General Comments: The manuscript provides novel research that has important implications in the management of an important scarab pest. I did not have much of an issue with the research performed, the analysis performed, or the conclusions reached. The main issue was the clarity of writing and understanding details in the methods, results and discussion. Because of some of the clarity issues, it was difficult for me at times to follow what the authors were doing and to understand their concepts about scarab abundance, sex ratio, etc. I think writing clarity is the single biggest issue with the manuscript, but some of the suggestions I have made in track changes may help with clarity.
I did find a few mistakes (all indicated in track changes) that absolutely must be changed (e.g., misspellings, not giving full scientific name at first use including naming authority, apparently incorrect author name for C. costelytra which should be (White) not White, and a possible mis-labeling of the sites in the right and left figures of Fig. 4, a missing reference citation, and a mistake on the Reinecke et al. 2002 citation [see manuscript for explanation]).
The authors have elevated the subfamily Melolonthinae to the family level of Melolonthidae, but I can find no taxonomic record of this subfamily being changed. If it has been elevated from subfamily to family, please provide support for that. Otherwise, use the correct Scarabaeidae.
The authors also appear to be using European/British English in a few places like behaviour rather than behavior, generalised rather than generalized, analysed rather than analyzed, etc. I have marked most of these and the authors need to consult with the journal editor about which English style is preferred by PeerJ and then do a global search to change as appropriate. Overall, I would recommend acceptance with revisions to correct clarity and improve understanding.

Annotated reviews are not available for download in order to protect the identity of reviewers who chose to remain anonymous.

Reviewer 2 ·

Basic reporting

The manuscript is conceptually sound, however the authors failed to provide important details regarding the arrangement of plants (grapes) within and between rows, plant height and width (spread) in the two commercial vineyards. This information is critical to assessing and interpreting the results of the work performed. If such information could be incorporated, it would dramatically improve the manuscript. In its current state, the manuscript should not be published. However, if improved, the manuscript will likely be acceptable.

Experimental design

No detail of the arrangement of plants (grapes) within and between rows, plant height and width (spread) in the two commercial vineyards. Such information is critical to interpreting the results of the data collected as these variables will likely have a profound effect.

Validity of the findings

Because there is a lack of specific information that will likely influence the interpretation of the results, it is not possible to determine the validity of the findings.

Additional comments

It is suggested that the authors incorporate the missing information and re-submit the manuscript for further review.

Reviewer 3 ·

Basic reporting

The manuscript is well written, clearly presented and thorough. The introduction and supportive literature is adequate. One suggestion I have to improve clarity is to refer to the crop as grapevines instead of vines. Locally grapevine may be commonly referred to as vines, but for your international readers it was initially confusing knowing to what sort of vines the authors were referring. The figures were easy to interpret and supportive of the discussion.

I have a slight issue with the statement in lines 67-71 regarding pesticide use. The authors state that insecticides are used prophylactically and perhaps this is the case in some crops such as tamarillo vines, but I think the authors need to specifically address procedures to protect grapevines. There is a very good outreach bulletin provided by New Zealand Winegrowers (2013), titled Grass Grub Beetle Damage http://www.nzwine.com/assets/sm/upload/5t/mi/n6/l5/NZPD100%20Grass%20Grub%20Beetle%20Damage%20FINAL2.pdf, that explains managing adult beetles in grapevines. In that publication they do not prescribe prophylactic insecticide applications, but applications based on adult trap monitoring and assessing crop damage. Essentially good IPM practices. Additionally, the authors then go on to state that such an approach, I am assuming the prophylactic one, is strongly discouraged. If this approach is not the norm in grapes, then this statement is misleading as to what actually occurs in grapevine IPM in New Zealand.

Experimental design

There are no issues with the experimental design and statistical analyses. I think the authors had a nice approach to how they went about collecting and handling the data.

Validity of the findings

I think the information reported as it pertains sex ratios is supported by the data, although I would suggest delving a little further into the possible reasons for the differences in sex ratios between Awatere and Blenheim. The authors state that the difference may be due to environmental conditions but do not explain how or what these conditions were or how they may have influenced the ratio. The authors state that at 47% male in 2015 at Awatere that the sample was female biased. Please provide standard error for these means. If the SEM includes 50% you cannot really conclude that the sample was female biased. Additionally, I can see where the differences in sex ratio from 1:1 is simply an artifact of sampling error.

Similarly, on lines 262-263, the authors state that the differences in adult abundance and flight at dusk between locations may be due to differences in sample sizes. This is probably the case, and infers sampling error, but I think it needs to be clarified as such.

There is a great deal of discussion (line 265-311) in the manuscript regarding female beetle behavior, pheromone attraction and preference for landing on tall and pronounced plant silhouettes and how these factors may be employed to aid in pest management. Although the discussion is a nice overview of the situation, it really has very little to do with the scope of the paper as related to the data. Similarly, the title of the manuscript is highly misleading. Upon reading the title I expected to encounter information pertaining to how sex differences may influence pest management decision making. None of the data support this. The previously mentioned outreach bulletin New Zealand Winegrowers (2013), covers this information. Thus what is provided in the manuscript is not speculative assessment of what may be a technique to manage these beetles in grapevines, but simply a reiteration of something already published. In my opinion the entire portion of the manuscript addressing pest management should be struck or reduced greatly. The data presented does not address pest management, but is a nice piece of work regarding sex ratios. Avoid stretching the manuscript into something it is not. This only dilutes what is important and based on the data provided.

Additional comments

I greatly appreciate the time taken in the preparation of the manuscript. Very well written and easy to review.

---

## Round 0.2 · accepted · Accept

The authors have done a good job of responding to the three reviewers' comments and have sufficiently revised or rebutted each request. Thanks to the reviewers and the authors for their work on this MS.

I notice that while the authors response to Reviewer 1's taxonomic comment was to revise their MS to Scarabaeidae: Melolonthinae, which is, I believe, correct. However within the MS the authors still refer to the beetles in this study as melolonthids, when by the correct taxonomy they should be melolonthines. I encourage the authors to correct this throughout while in production.

There was some discussion over the use of "British" or "American" English. PeerJ will accept either as long as it is used consistently within a MS.

It would be helpful if the authors would elect to publish the review history alongside their paper, as it provides some valuable information for the historical record, and for pest managers who may be evaluating these data in terms of utility in an overall IPM strategy.